# A Phase II Study on the Effect of Taurisolo^®^ Administered via AEROsol in Hospitalized Patients with Mild to Moderate COVID-19 Pneumonia: The TAEROVID-19 Study

**DOI:** 10.3390/cells11091499

**Published:** 2022-04-29

**Authors:** Stefano Sanduzzi Zamparelli, Ludovica Capitelli, Nicola Coppola, Claudia Venditto, Ciro Santoro, Giuseppe Annunziata, Dario Bruzzese, Nunzia Cuomo, Ivan Gentile, Marialuisa Bocchino, Alessandro Sanduzzi Zamparelli

**Affiliations:** 1Pneumology I and Respiratory Pathophysiology Unit, Department of Translational Medical Sciences, University of Campania “Luigi Vanvitelli”, A. Cardarelli Hospital, 80131 Naples, Italy; stefanosanduzzi@gmail.com; 2Pneumology I and Respiratory Pathophysiology Unit, Department of Clinical Medicine and Surgery, Section of Respiratory Disease, University Federico II, A. Cardarelli Hospital, 80131 Naples, Italy; ludovica.capitelli@gmail.com; 3Infectious Diseases Unit, Department of Mental Health and Public Medicine, University of Campania “Luigi Vanvitelli”, 80131 Naples, Italy; nicola.coppola@unicampania.it; 4Department of Clinical Medicine and Surgery, Section of Respiratory Diseases, University Federico II, Azienda Ospedaliera dei Colli-Monaldi Hospital, 80131 Naples, Italy; claudia.venditto89@gmail.com (C.V.); marialuisa.bocchino@unina.it (M.B.); 5Department of Advanced Biomedical Sciences, University Federico II, 80131 Naples, Italy; ciro.santoro@unina.it; 6Department of Pharmacy, University Federico II, 80131 Naples, Italy; giuseppe.annunziata@unina.it; 7Department of Public Health, University of Naples Federico II, 80131 Naples, Italy; dbruzzes@unina.it; 8Microbiology and Virology Unit, Cotugno Hospital, 80131 Naples, Italy; nunzia.cuomo@ospedalideicolli.it; 9Department of Clinical Medicine and Surgery-Infectious Diseases, University of Naples Federico II, 80131 Naples, Italy; ivan.gentile@unina.it; 10Staff of UNESCO, Chair for Health Education and Sustainable Development, University Federico II, 80131 Naples, Italy

**Keywords:** polyphenols, COVID-19 pneumonia, ICU access

## Abstract

Background: Polyphenols are the largest class of bioactive compounds in plants, which are synthesized as secondary metabolites. In the last few years, interesting studies have demonstrated the efficacy of polyphenols against coronavirus infections. Methods: we conducted a phase II multicentric clinical trial (TAEROVID-19) during the first wave of the COVID-19 pandemic in order to assess the safety and feasibility of Taurisolo^®^ aerosol formulation in hospitalized patients suffering from SARS-CoV-2 pneumonia. Results: we observed a rapid decline of symptoms and a low rate of intensive care in patients treated with Taurisolo^®^, with a faster decline of symptoms. Conclusions: This is the first trial assessing the safety and feasibility of Taurisolo^®^ aerosol formulation. We could argue that this treatment could act as an add-on therapy in the treatment of COVID-19 patients, owing to both its anti-inflammatory and antioxidant effects. Further controlled trials are needed, which may be of interest to evaluate the compound’s efficacy.

## 1. Introduction

The SARS-CoV-2-related epidemic, firstly reported in December 2019 in Wuhan, China, rapidly spread worldwide, becoming a pandemic, as defined by the World Health Organization (WHO) in March 2020 [1,2]. Nowadays, this pandemic is still responsible for high morbidity and mortality rates, resulting in a dramatic socio-economic issue. The diagnosis of SARS-CoV2 infection is based on qualitative real-time reverse transcriptase–polymerase chain reaction (rRT-PCR) analysis on a nasopharyngeal swab [3].

SARS-CoV2 infection has a varying clinical course, including asymptomatic or pauci-symptomatic patterns or, in many cases, causing severe systemic complications. According to data from the first and second wave, about 80% of patients are pauci-symptomatic or asymptomatic, while 15% show severe patterns with pneumonia and other complications, and 5% suffer from critical illness [4]. The first stage of the infection is characterized by viral replication (stage I), followed by inflammation, mainly at the level of the respiratory system (stage II), that, in the most severe cases, may result in a dramatic septic-like scenario [5]. Later, there is a hyperactivation of the immune system with a release of inflammatory cytokines (IL-1, IL-6, TNF-α, etc.) from macrophages and endothelial cells, and the release of other inflammation mediators (D dimer, LDH, ferritin, CRP, etc.), in a clinical situation similar to the “cytokine storm syndrome” [5]. Pathogenesis of severe pneumonia and respiratory distress, indeed, is characterized by diffuse damage at the endothelial, capillary, and alveolar levels [6] that results in increased release of inflammatory cytokines with consequent pulmonary edema and hyaline membrane formation [7]. Epidemiological studies conducted during the first wave of COVID-19 hospitalized patients reported an ~15–28% mortality rate, while about 15–26% of patients need intensive care [8,9]. Among these last, mortality is about 50% [10].

During the first wave, before the anti-SARS-CoV-2 vaccination, a variety of therapeutic approaches were tested, such as chloroquine, idroxychloroquine, azithromycin, lopinavir–ritonavir, favipiravir, ribavirin, interferon, remdesivir, steroids, and anti-IL6 inhibitors [11,12,13,14,15], while only these latter three have been demonstrated to improve clinical outcomes in specific populations.

Polyphenols are the largest class of bioactive compounds in plants, which are synthesized as secondary metabolites. In the last few years, interesting studies demonstrated the efficacy of polyphenols against coronavirus infections. These are in vitro studies conducted on different cell models infected with micro-organisms belonging to the coronavirus family. In general, these studies reported that in vitro polyphenols exert a marked and well-demonstrated activity against coronavirus [16]. All these studies conclude that the main mechanisms of action are related to the reduction of viral load and inhibition of nucleocapsid protein expression [17]. Additionally, the novel nutraceutical formulation containing grape pomace extract in humans (Taurisolo^®^) has relevant antioxidant and anti-inflammatory activities in humans, in addition to a potential antiviral efficacy [18,19,20,21,22]. The delivery of pharmacological compounds toward the lungs, theoretically, is the best way to treat respiratory diseases [23]. Inhalation, indeed, allows the drug to more rapidly and directly reach the site of action in the pulmonary tissue; moreover, reducing the amount of drug to administer and, consequently, the side effects. These advantages are particularly relevant for polyphenols that, in turn, are generally characterized by low bioavailability due to rapid metabolism and low water solubility [24].

For all these reasons, we conducted a phase II multicentric clinical trial (TAEROVID-19) during the first wave of the COVID-19 pandemic in order to assess the safety and efficacy of Taurisolo^®^ aerosol formulation in hospitalized patients suffering from SARS-CoV-2 pneumonia.

## 2. Materials and Methods

### 2.1. Study Design and Participants

TAEROVID-19 is an Investigator-Initiated Study (IIS), a phase II clinical trial, multicentric (Azienda Ospedaliera dei Colli, Infectious diseases unit, University Federico II, Infectious disease unit University of Campania “Luigi Vanvitelli”, Naples, Italy), single arm, aimed mainly at evaluating the safety and efficacy of Taurisolo^®^-based aerosol formulation on admission to intensive care units (ICU) in hospitalized patients with COVID-19-related pneumonia as an add-on therapy. All participants provided written informed consent before study participation based on the principles of the Declaration of Helsinki. The study protocol and amendments were approved by the institutional review board and independent ethics committees at each center (local ethical committee: ref AOC/0013770/2020); the study was conducted according to Good Clinical Practice guidelines defined by the International Council for Harmonisation and local laws. The primary endpoint was the rate of ICU admission (based on very low P/F ratio, and/or multi-organ failure, and/or other clinical condition that requires invasive mechanical ventilation) or death.

Eligible patients were aged at least 18 years old with SARS-CoV-2 infection (confirmed by nasopharyngeal swab test), lung involvement at admission proved by HRCT (high-resolution computed tomography) findings ranging between 10/20 and 16/20 of Chung score, and lung function mildly impaired (P/F ratio > 150), consistent with the definition of mild to moderate COVID-19 pneumonia.

Patients were ineligible to be included in the study if requiring invasive mechanical ventilation at admission (unlike patients requiring non-invasive mechanical ventilation are eligible), requiring vasopressors or inotropes, having symptom onset more than 7 days from admission, having a secondary bacterial and/or mycotic infection, having already started a treatment with IL6 inhibitors (e.g., Tocilizumab), or with a prognosis of less than 24 h (imminent death).

Patients received a nebulization of Taurisolo^®^ (100 mg, containing 0.56 mg catechins) 3 times per day (every 8 h) and Polygonum cuspidatum E.S (95% resveratrol, 4.75 mg) administered in 10 cc of physiological saline solution by Portable YM-252 Mesh Nebulizer whit CPAP oronasal mask AirFit F20 (ResMed). Patients were treated with study medication for a maximum of 14 days, until informed consent was withdrawn, or until hospital discharge or ICU admission or death.

Eligible and consenting patients were studied by collecting data about clinical, radiological, and epidemiologic data such as radiological score [25] (obtained from HRCT findings like ground glass, fibrosis, consolidation, pleural effusion, subpleural thickenings), age, anthropometric data, cardiovascular risk factors (hypertension, diabetes, obesity, etc.), comorbidities (kidney failure, oncologic history, etc.), previous home therapy, concurrent treatment for COVID-19, oxygen-therapy demand, FiO2, body temperature, respiratory rate, heart rate, blood pressure, pulse oximeter blood oxygen percentage and symptoms (cough, headache, malaise, dyspnea, etc.).

Our cohort was followed up by collecting baseline laboratory data (blood samples were collected at baseline, on day 7 and 14 and stored at −20°C) as blood gasses analysis (SpO2, PCO2, PaO2, pH, P/F, lactate, etc.), blood count, routine lab test, (creatinine, blood urea, transaminase, etc.), and inflammatory markers (CPR, ESR, pro-calcitonin, D-dimer, fibrinogen, IL-6, IL1, TNF-α, KL-6).

A nasopharyngeal swab was collected at baseline (T0), at day 1 (T1), 3 (T2), 5 (T3), 7, 14, 21, and 28 with viral load measurement through Cycling threshold (Ct). A real-time polymerase chain reaction (RT-PCR) Ct, analyzed from respiratory samples taken by swab, is a semiquantitative measure of viral load widely used to monitor SARS-CoV-2 infection. Commonly, the viral genes detected include nucleocapsid (N) and envelope (E) proteins that play a key role in viral self-assembly, RNA-dependent RNA polymerase (RdRP), and spike (S) glycoprotein that interacts with the host cell’s ACE2 receptors.

The Ct value is inversely proportional to the amount of viral nucleic acid in specimens (meaning that a lower Ct value indicates a higher amount of virus) and can be used to estimate viral load.

The compliance of the therapy was evaluated by considering the number and duration of dispensed therapies according to the following classification: optimal (70% of the planned treatment dispensed), good (between 30–70% of the planned treatment), poor (less than 30% of the therapy is delivered).

### 2.2. Taurisolo Preparation

Taurisolo^®^ is a polyphenolic extract from Aglianico cultivar grapes, collected during the autumn 2018 harvest, produced at a large scale by the MB-Med Company (Turin, Italy). Briefly, grapes were extracted with hot water (50 °C); the obtained solution was filtrated, concentrated, and spray-dried, obtaining a fine powder. For the formulation used in the trial, Polygonum cuspidatum extract containing 98% resveratrol was added to Taurisolo^®^. The polyphenol profile was evaluated by a high-performance liquid chromatography-diode array detector (HPLC-DAD, Jasco Inc., Easton, MD, USA) analysis using the method described by Giusti et al. [26] and previously reported [27,28]. The HPLC-DAD analysis was performed on Taurisolo^®^ + Polygonum cuspidatum formulation before and after aerosol. More specifically, during aerosol, the nebulized formulation was collected through a plastic pipe connected to the device and condensed into a falcon tube. The solution was then lyophilized and solubilized in methanol for the HPLC-DAD analysis. The polyphenol profile of the Taurisolo^®^ + Polygonum cuspidatum formulation before and after aerosol is reported in Table 1. Mean values reported refer to a single dose (100 mg Taurisolo^®^ + 4.75 mg Polygonum cuspidatum) of the formulation used in the trial.

### 2.3. Safety Analysis

Safety was assessed through the description of adverse events (AE) observed on the EP. These were reported individually, and if a patient presented the same AE repeatedly during the follow-up course, only the most severe was reported. Descriptive tables with the incidence of the main AEs were therefore constructed.

Aerosol therapy was considered (e.g., intubation, bronchoalveolar lavage. and oxygen therapy with high-flow nasal cannula) a high-risk procedure for aerosolization and virus dispersion. However, there are several measurements to make this risk negligible: (i) aerial isolation of the patient being treated and use of personal protective equipment as required by current recommendations on hospitalized patients; (ii) use of CPAP masks that reduce the dispersion of exhaled material. The latter point was demonstrated through a test performed at the Pharmacy Department of Federico II University. The protocol applied for this evaluation is shown below.

The safety of the YM-252 (Mesh Nebulizer) and oronasal mask for CPAP AirFit F20 (ResMed, San Diego, CA, USA) misting system has been tested in humans to evaluate the risk of aerosolization of the virus. For this purpose, the mask was connected to the aerosol device by a collector. Some strips of absorbent paper (n4, 20 cm each) have been applied to the outside of the mask, along the entire external perimeter in contact with the face, in order to create a sort of breathable seal.

Folin-Ciocalteu reagent is a commercially available reagent, based on a mixture of aqueous solution and phosphomolybdate and ammonium phosphotungstate, used in analytical chemistry for the determination of phenols and polyphenols. The reduction of the reagent in an alkaline environment leads to the formation of anions, which are intensely colored blue.

The reagent is sprayed on the paper strips by a special nebulizer at the end of the aerosol application. The possible presence of polyphenols on the paper is detected by the appearance of blue spots.

### 2.4. Statistical Analysis

Sample size was based on a Fleming–A’Hern design. Based on the current evidence, it was assumed that a rate for the primary outcome ≥10% would be too larger to warrant future trials but that a rate ≤1% would warrant future investigations. Using an alpha level of 0.05 and a beta level equal to 0.10, the target recruitment was equal to 46 patients with zero or only one event to be observed for further research to be considered feasible. This was the smallest number that satisfied the design criteria. However, the target sample size was not reached, and only 43 patients were evaluable for analysis. Therefore, the actual alpha level was equal to 0.062 with a power of 0.931.

The study population evaluated to test the efficacy for primary and secondary endpoints consisted of all patients enrolled in the study who received at least one dose of the experimental treatment (EP: Evaluable Population/Per-protocol analysis). The safety analysis was also being conducted on the EP.

Standard descriptive analyses were used to describe the characteristics of the enrolled sample. Average, standard deviation (SD), median, minimum, and maximum were calculated for the numeric variables, while the categorical variables were described by absolute frequencies and percentages. The admission rate in ICU was obtained by dividing the number of subjects who underwent admission to the ICU by the total of the EP. The corresponding upper end of the 90% confidence interval was calculated for the formal verification of the hypothesis that the real admission rate to the ICU was lower than 1%.

The change in the cumulative incidence of events at the end of the follow-up was also calculated. The longitudinal trajectories of the laboratory parameters and clinical data were assessed using mixed linear models (with intercept and random angular coefficient). The time was entered as a numeric variable. No interim analyses were planned. Any subgroup analyses were considered exploratory as the study was not sized for a comparison of effectiveness between different groups.

## 3. Results

Forty-three patients (32 males, 11 females) were recruited for the trial with a mean age of 61.8 ± 12.4 years (range: 20.9 to 84.5). The target sample size was not reached, and only 43 patients were evaluable for analysis. As things stand, the actual alpha level was equal to 0.062 with a power of 0.931. Mean BMI was equal to 27.9 ± 4.4 (range: 17.7 to 40.1). Thirteen patients (30.3%) were either actual smokers (*n* = 6; 14%) or former smokers (*n* = 7; 16.3%). Most frequent comorbidities at baseline were hypertension (*n* = 24; 58.5%) and diabetes (*n* = 7; 17.1%) (Table 2) while steroid drugs and gastroprotective agents were the most frequently used drugs (Table 3).

On hospital admission the most prevalent COVID-19-related symptoms were fever (*n* = 30; 76.9%), dyspnea (*n* = 30; 76.9%), and cough (*n* = 29; 74.45) (Figure 1).

All patients received at least one dose of the experimental treatment with a median treatment duration of 7 days (range: 1 to 14 days) as an add-on therapy, in addition to the usual therapeutic approaches summarized in Table 4. All eligible patients were treated with antiviral therapy; patients with reduced mobility practiced LMWH therapy at prophylaxis dosage; one patient was subjected to LMWH therapy at anticoagulant dosage for the presence of pulmonary embolism; one patient continued anticoagulant therapy previously undertaken for atrial fibrillation. Patients with extended presence of ground glass opacity to the chest CT were subjected to high-dose steroid therapy. Antibiotics were introduced in patients with procalcitonin >0.5 ng/mL.

During a median follow up of 10.5 days (range: 1 to 31 days), only one patient required ICU admission, thus rejecting the null hypothesis that the ICU admission rate was equal or higher than 10% in favor of the alternative hypothesis that the rate was equal or lower than 1%. The estimated rate was equal to 2.33% (95% one-sided confidence interval: 0 to 9.77%).

During the first seven days of treatment, a significant reduction in several inflammatory markers was observed. In particular, median CPR decreased from 8.8 (IQR: 4.7 to 17) to 0.5 (IQR: 0.4 to 2) (*p* < 0.001) and median IL6 decreased from 22 (IQR: 8.8; 40) to 4.3 (IQR: 3; 8.9) (*p* < 0.001). Furthermore, fibrinogen levels lessened from 585 (IQR: 498; 687) to 377 (IQR: 325; 462) (*p* < 0.001). No significant reduction was observed for D-Dimer or white blood count (WBC) (Figure 2).

Virological data, measured by evaluation of Ct from baseline to day 5, showed a faster decreasing trend, although not enough to reach statistical significance (Table 5). At T0 (baseline), the mean of C (t) was 29.1 with a sample of 45 patients. At T1 (day 1), the mean of C (t) reached 32.8, with a sample of 37 patients. At T2 (day 3), the mean of C (t) reached 34.3 with a sample of 28 patients. At T3 (day 5), the mean of C (t) was 35.4 with a sample of 18 patients. We found that in all patients, independently from remdesivir therapy, there was a faster increase in C (t) within a few days—this correlates with a more rapid negativization of the patient and, therefore, with lower contagiousness.

P/F ratio increased significantly during the first seven days of treatment, rising from 292 ± 60.4 to 310 ± 73 (*p* = 0.033) as well as oxygen saturation (95.8 ± 2 at baseline vs. 97.1 ± 1.6 at day 7; *p* < 0.001). On the contrary, the observed increase in PaO2 did not reach statistical significance (72.8 ± 14 vs. 77.6 ± 11.4; *p* = 0.083) (Figure 3).

Because radiological improvement was slower than clinical, we did not perform a second HRCT during hospitalization.

Safety analysis was performed on all patients. Only one adverse event was reported in a patient with bronchial hyperreactivity (cough not controlled by sedatives), related to the mode of administration of the drug. Only one patient discontinued treatment early due to worsening respiratory failure (later died in the ICU). There was only one death in a patient with severe respiratory insufficiency, who died after being transferred to the intensive care unit because of a concomitant massive pulmonary embolism, with no radiological signs of deep vein thrombosis.

## 4. Discussion

This nutraceutical proved to be very safe, in terms of the occurrence of adverse effects, since only one patient out of 43 was obliged to discontinue the treatment owing to unrestrained cough related to bronchial hyperreactivity. Moreover, aerosol therapy is considered, like intubation, bronchoalveolar lavage, and oxygen therapy with high-flow nasal cannula, a high-risk procedure for aerosolization and potential viral dispersion. In our experience, conversely, we accurately demonstrated, by a colorimetric control test, that there was no dispersion.

In fact, this trial demonstrated that, even if in a small group of patients, topic administration of this drug via the aerosolic route could be considered safe in COVID-19 patients. A recent paper by Ramakrishnan et al. shows that inhaled glucocorticoid budesonide, given for a short duration, might be both an effective treatment of early COVID-19 in adults, and a safe method of administration, in terms of lack of viral spreading [29]. 

We found in our patients, independently from remdesivir therapy, a rapid negativization of the patient and, therefore, less contagiousness expressed in an increase in C (t) within a few days. It is interesting to analyze these data and evaluate those present in other experiences. As reported by Singanayagam et al., the duration of infectiousness following mild-to-moderate COVID-19, closely related to the level of SARS-CoV-2 RNA, declines over time progressively, with a probability of culturing virus that declined to 6% at 10 days after symptom onset [30].

Xiaowen et al. underline that negative conversion of SARS-CoV-2 RNA takes 14 days on average (IQR: 10–18) from the first positive RT-PCR test. In this study, they found that the rate of RNA negative conversion within 7 days, 14 days, and 21 days among all patients were 10.2% (95% confidence interval, 95%CI: 2.1–17.5%), 62.7% (48.1–73.2%), and 91.2% (80.4–96.4%), respectively [31].

Carini et al. found that the time length of negativization, calculated from the beginning of symptom onset to two consecutively negative RT-PCR test results, collected at least 24 h apart after the disappearance of symptoms, required three weeks or more in the majority of subjects. More than 70% of symptomatic patients became test-negative within four weeks; this proportion rose to 80% in the asymptomatic ones. Moreover, 29.2% of symptomatic versus 20% of asymptomatic patients became test-negative at five weeks [32]. Those findings are supported by the Zhou study, in which SARS-CoV-2 was detected in respiratory samples for a median of 20 days, up to 37 days after symptom onset. Of all results from clinical trials on Remdesivir, only one publication reported on its impact on SARS-CoV-2 viral load [33].

Furthermore, the British National Institute for Health and Care Excellence (NICE) states that the aerosols produced by the patients affected by SARS-CoV-2 infections are generated from the fluid within the nebulizer chamber that does not carry the patient–viral derived particles [34]. In fact, if a particle in the aerosol comes into contact with the contaminated mucous membrane, it ceases to be dispersed in the air and, therefore, will not be aerosolized. However, it is essential to pay particular attention (wash hands and don clean gloves) when removing the nebulizers.

In our study, we observed a low rate of ICU admission (based on P/F ratio below 150 and/or multi-organ failure and/or other clinical condition that requires invasive mechanical ventilation) in contrast to other authors’ experience since only one patient out of 43 (2.3%) required ICU admission. Colaneri et al. report an access rate to ICU of 6.8%, and Zhang and coworkers report a rate ranging from 6.5 to 13.7%, according to age [35,36]. Furthermore, several inflammatory markers (IL-6, CRP, fibrinogen) showed a significant reduction in serum levels after seven days of treatment, but it should be emphasized that it is impossible to argue if such results could depend on the standard of care or the real effect of Taurisolo^®^. Similar results were also shown in a paper by Mahat et al., who described a slower decline in serum levels of inflammatory markers in mild to moderate COVID-19 pneumonia [37]. Our results also highlighted a significant improvement in pulmonary function in particular during the first 7 days of treatment with Taurisolo^®^, showing an increase in P/F ratio from 292 ± 60.4 at baseline to 310 ± 73 at day 7 (*p* = 0.033) and oxygen saturation from 95.8 ± 2 at baseline vs. 97.1 ± 1.6 at day 7 (*p* < 0.001). Comparing our results with what is reported in the literature, a faster decline of symptoms and a lower rate of ICU access in patients treated with Taurisolo^®^ [4] were observed.

Besides, a recent paper by Xi-Jian Dai et al. underlines that consumption of red wine played protective effects against the COVID-19, while consumption of beer and cider increased the COVID-19 risk: such a result could strengthen the potential anti-inflammatory and antioxidant utility of Taurisolo^®^, which contains grape pomace extract [38].

Our trial began at a time when COVID vaccines were not yet available. During the study, the first RNA vaccines were distributed to healthcare professionals, but such protection was not yet available to the general population. None of our enrolled patients had access to vaccination before contracting COVID-19 pneumonia. We believe in a possible “booster” role of Taurisolo^®^, in addition to vaccination.

From the literature, we know that grape pomace extract (GPE) has antiviral activity against various micro-organisms. Several mechanisms of action have been demonstrated, including down-regulation of virus entry co-receptor expression, suppression of virus replication, and reduction of induced inflammation. In addition to its direct antiviral activity, GPE was found to be effective in alleviating the pathological complications of respiratory viral infection by reducing the expression of pro-inflammatory mucins and interleukins, including IL-1β, −6, and −8 [38].

Most observational studies assessed that the mRNA vaccines against COVID appear to be safe and highly effective tools to prevent severe disease, hospitalization, and death against all variants of concern. The protection appears to be against symptomatic and asymptomatic infection. mRNA vaccines were associated with a faster decline in viral load against several variants, including Delta, and a lower probability of viral culture positivity [39,40].

Based on these considerations, we hypothesize a future use of Taurisolo^®^ in reducing the inflammatory cascade and, therefore, the severity of the disease, in addition to the vaccines, given the negligible risk profile related to its nebulization compared to the possible benefits.

Our trial suffers from some limitations: first of all, the small number of patients; then, the absence of a control group, even if it must be considered that this is a phase 2, single-arm uncontrolled study, and therefore represents a proof of concept. The lack of a control group is mainly linked to ethical reasons, in relation to the need to guarantee, especially during the first wave of COVID-19, the best possible treatment for the entire population under study, without giving up drugs that could alter the validity of our data (corticosteroids, antiviral drugs, monoclonal antibodies). Finally, the dynamic changes of the pandemic through time and the development of different variants of SARS-CoV-2 limits the extrapolation of those results on the current infected population; however, we think that Taurisolo^®^ nebulization could represent a solid adjuvant to reduce the time of infection and contagiousness along with vaccination, in order to reduce the spreading of the disease.

## 5. Conclusions

In conclusion, we could argue that aerosol therapy with Taurisolo^®^ could act as an add-on therapy in the treatment of COVID-19 patients, owing to both its anti-inflammatory and antioxidant effects [39]. Besides, because of the frequent persistence of COVID in nasopharyngeal swabs, Taurisolo^®^ could be used in paucisymptomatic non-hospitalized patients to accelerate negativization. Therefore, in the future, such a nutraceutical could be useful in the therapy of chronic pulmonary diseases such as COPD, bronchial asthma, and bronchiectasis.

## Figures and Tables

**Figure 1 cells-11-01499-f001:**
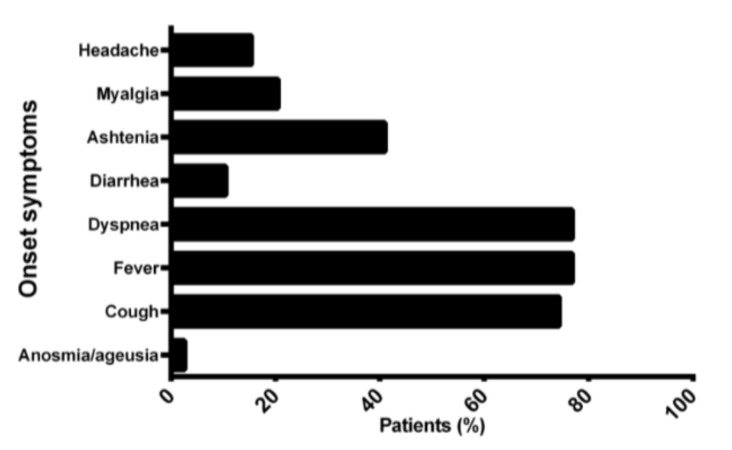
Most prevalent symptoms at COVID-19 pneumonia onset. Data are expressed as the percentage of patients.

**Figure 2 cells-11-01499-f002:**
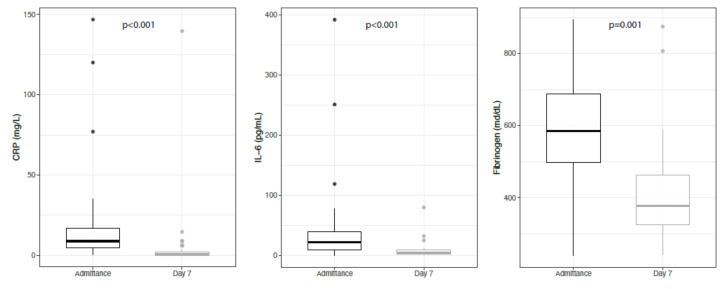
Inflammatory parameters at the admission and at day seven after treatment.

**Figure 3 cells-11-01499-f003:**
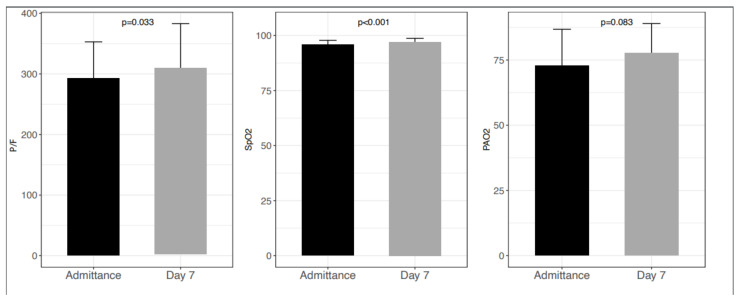
Lung function parameters. P/F ratio increased significantly during the first seven days of treatment, rising from 292 ± 60.4 to 310 ± 73 (*p* = 0.033) as well as oxygen saturation (95.8 ± 2 at baseline vs. 97.1 ± 1.6 at day 7; *p* < 0.001). On the contrary, the observed increase in PAO2 did not reach statistical significance (72.8 ± 14 vs. 77.6 ± 11.4; *p* = 0.083).

**Table 1 cells-11-01499-t001:** High-performance liquid chromatography-diode-array detector (HPLC-DAD) analysis of the main polyphenols contained in the formulation Taurisolo^®^ + Polygonum cuspidatum, before and after aerosol. Values are expressed in µg per single dose of Taurisolo^®^ 100 mg + Polygonum cuspidatum 4.75 mg ± standard deviation of three repetitions.

Compound	Mean Value (µg/Taurisolo^®^ 100 mg + *Polygonum* *Cuspidatum* 4.75 mg sin gle do se) ± SD	
	Pre-Aerosolization	After Aerosolization
Ferulic acid	1.459 ± 0.098	n.d.
Resveratrol	4656.25 ± 21.00	3597 ± 38.00
Caffeic acid	3.50 ± 0.300	3.273 ± 0.214
*p*-cumaric acid	12.275 ± 0.277	0.47 ± 0.122
Rutin	9.881 ± 0.731	n.d.
Quercetin	13.541 ± 0.469	11.192 ± 0.234
Procyanidin B1 dimer	94.633 ± 5.520	30.553 ± 1.842
Procyanidin B2 dimer	64.589 ± 5.917	18.155 ± 2.500
Syringic acid	31.095 ± 0.001	29.101 ± 1.842
Epicatechin	169.655 ± 10.960	26.157 ± 0.443
Gallic acid	19.946 ± 0.459	n.d.
Catechin	249.904 ± 30.741	61.662 ± 0.353

**Table 2 cells-11-01499-t002:** Clinical data at admission.

Clinical Data at Admission (43 Patients)	
Mean HRCT Chung score	12/20 [10/20; 16/20]
Mean P/F	290 [206; 434]
Mean lactates (mmol/L)	1.17 [0.4; 2.8]
Mean Procalcitonin (ng/mL)	0.13 [0.03; 1.6]
Mean CRP (mg/L)	17.54 [0.5; 143]
Mean IL-6 (pg/mL)	44.8 [2.5; 390]
Mean KL-6	341.16 [203; 711]
Mean white cells	207.5 [133.2; 325.2]
Mean Dimer	585 [498; 687]
Mean Fibrinogen	6.2 [4.2; 9.9]

**Table 3 cells-11-01499-t003:** Baseline characteristics of the study population.

Baseline Characteristics of Patients	
Mean age	61.8±12.4 years
Mean BMI	27.9 ± 4.4Kg/m^2^
Sex	
Female	11 (25.5%)
Male	32 (74.5%)
Smoke	
Actual smokers	6 (14%)
Former smokers	7(16.3%)
Never smokers	30 (69.7%)
Comorbidities	
Immunosuppression	3 (7.5%)
COPD	3 (7.5%)
Asthma	3 (7.5%)
Interstitial lung disease	1 (2.5%)
Bronchiectasis	2 (5%)
Hepatopathy	1 (2.5%)
Arterial hypertension	24 (58.5%)
Dyslipidemia	4 (10.3%)
Myocardial infarction	5 (12.8%)
Hypertensive heart disease	2 (5%)
Ischemic heart disease	5 (12.8%)
Atrial fibrillation	3 (7.7%)
Arterial vasculopathy	2 (5.1%)
Active solid neoplasia	3 (7.7%)
Diabetes	7 (17.1%)
Previous home therapy	
Oral Steroids	13 (33.3%)
PPI (proton pump inhibitors)	12 (31.6%)
ICS/LABA	2 (5.3%)
ICS/LABA/LAMA	1 (2.6%)
LAMA	2 (5.1%)
Oxygen therapy	2 (5.1%)
ACEi (ACE inhibitors)	9 (23.1%)
Angiotensin receptor antagonists	9 (23.1%)
ASA (acetylsalicylic acid)	6 (15.4%)
Other antiplatelets	1 (2.6%)
DOAC (direct oral anticoagulants)	4 (10.3%)
Insulin	2 (5.1%)
Oral hypoglycemics	2 (5.1%)

**Table 4 cells-11-01499-t004:** COVID-19 pneumonia treatment.

COVID-19 Pneumonia Treatment	
Remdesivir	9 (20%)
High-dose corticosteroids	19 (44%)
Low-flow oxygen therapy	8 (18.6%)
High-flow oxygen therapy	5 (11.6%)
Low molecular weight heparin as prophylaxis (LMWH)	14 (32.6%)
Low molecular weight heparin as therapy (LMWH)	1 (2.3%)
Other anticoagulants	1 (2.3%)
Antibiotics	10 (23.3%)

**Table 5 cells-11-01499-t005:** Viral charge expressed in c (t) in patients treated with Taurisolo^®^ at baseline and days of treatment.

	Day 0	Day 1	Day 3	Day 5
Taurisolo^®^	29.1 (43 pt)	32.8 (37 pt)	34.3 (28 pt)	35.4 (18 pt)
Tauri with Remde	30.2 (9 pt)	32.5 (9 pt)	34.9 (9 pt)	35.4 (9 pt)
Tauri without Remde	30.2 (34 pt)	26.7 (28 pt)	29.7 (17 pt)	29.9 (9 pt)

## Data Availability

Not applicable.

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
