# Peer review of "A Phase II Study on the Effect of Taurisolo® Administered via AEROsol in Hospitalized Patients with Mild to Moderate COVID-19 Pneumonia: The TAEROVID-19 Study"

_cells, 2022, doi:10.3390/cells11091499_

Round 1

Reviewer 1 Report

The review is focused on a phase II study about the effect of Taurisolo® administered via 2 AEROsol in hospitalized patients with mild to moderate 3 COVID-19 pneumonia. The authors discussed based on their results the efficiency of the aerosol therapy with Taurisolo® could act as an add-on therapy in the treatment of COVID 19 patients. This will open the door for future research about the efficiency of Nutraceuticals. 

Strengths
It is a hot topic

Limitations
Small number of the cohort used.

Some graphs should be better presented 

Discussion about the effeciency of this kind of treatment with the availability now of many vaccines. I think this can be discussed. 

Author Response

Some graphs should be better presented

- We thank the reviewer for the valuable comments. We graphically revised the graphs as requested.

Discussion about the effeciency of this kind of treatment with the availability now of many vaccines. I think this can be discussed. 

- Thanks for your comment. We added a paragraph in the discussion section to discuss how the proposed treatment could be integrated as an option along with vaccination strategy

Reviewer 2 Report

This manuscript titled “A Phase II study on the effect of Taurisolo® administered via AEROsol in hospitalized patients with mild to moderate COVID-19 pneumonia: the TAEROVID-19 study” by Zamparelli et al., present the results of phase II study of aerosol containing taurisolo as active ingredient. The topic is interesting however there are so many points in this manuscript which needs to be corrected before it is being accepted for its publicatio.

How Taurisolo® was obtained? Was it commercially purchased or extracted from the plants. Authors should clarify this.

-If it was extracted from the plants they should give the detail of extraction procedure and modern chromatographic analysis results of the extracts using HPLC/GC-MS analysis results.

-if it was commercially purchased then they should give the detail of manufacturer and detail of active ingredients quantity.

Page 1 line 43.  antioxi-dant should be changed to antioxidant.

Page 12 line 373.  a/paucisyntomatic should be written as paucisymptomatic.

Page 12 line 373.  hospiatalized should be corrected as hospitalized.

Page 5 line 208-209. Figure 1 and 2 are not showing the difference for the detection of polyphenols. Authors should present the spectroscopic analsysis such as IR, NMR, MS for the detection of polyphenols.

Author Response

- How Taurisolo® was obtained? Was it commercially purchased or extracted from the plants. Authors should clarify this. -If it was extracted from the plants they should give the detail of extraction procedure and modern chromatographic analysis results of the extracts using HPLC/GC-MS analysis results. if it was commercially purchased then they should give the detail of manufacturer and detail of active ingredients quantity.

We agree with this reviewer in recognizing that more details about Taurisolo® and its composition should be included into the manuscript. Accordingly, we included in the Material and methods section the following paragraph “Taurisolo® preparation” where we described in detail the productive process as well as the extraction and composition.

- Page 1 line 43.  antioxi-dant should be changed to antioxidant.

Page 12 line 373.  a/paucisyntomatic should be written as paucisymptomatic.

Page 12 line 373.  hospiatalized should be corrected as hospitalized.

We modified the misspelled words in the text accordingly

- Page 5 line 208-209. Figure 1 and 2 are not showing the difference for the detection of polyphenols. Authors should present the spectroscopic analsysis such as IR, NMR, MS for the detection of polyphenols. ->

We thank the reviewer for the valuable comment. We removed figure 1 and 2, adding in supplemental table 1 data on polyphenol profile of Taurisolo® before and after aerosolization.

Reviewer 3 Report

Sanduzzi and colleagues evaluated the safety and efficay of Taurisolo iin patients admitted for COVID 19 pneumonia. This is a prospective uncontrolled therapeutic trial.

Major comments :

1-The authors compared their group of patients with historical controls. This cannot be accepted unless the control group is built to match the main charactristics of the patients in this study including the different treatments received, which is not the case here. I would suggest to remain in stable ground, just describe the population included and the outcomes. And add to the discussion section any comment about efficacy.   

2-There are paragraphs in the results section that should go in the discussion. For instance « Comparing our results… » or « As reported by Singanayagam… » and the next sentence « Xiawen et al…. » and also the sentence « Carini et al…. ».

3-The results section concerning the virological results is very difficult to read. It should not repeat the data which are already in the tables.

4-There was one death. This is a serious adverse event. Please correct the sentence in page 6 which states that there were no adverse events.

5-There are too many details in the tables. The authors should merge table 2 and 3 . Table 5 and 6 should be merged. In tables 5 and 6, change T0, T1, etc…to the relevant days : Day 0, day 1, day 3, day 5. This will help the reader. Table 7 is not a table ; it is a figure.

6-Figure 1 is useless. Should be deleted. Figure 3 and Table/Figure 7 need the SD bars  

Author Response

1-The authors compared their group of patients with historical controls. This cannot be accepted unless the control group is built to match the main charactristics of the patients in this study including the different treatments received, which is not the case here. I would suggest to remain in stable ground, just describe the population included and the outcomes. And add to the discussion section any comment about efficacy. ->

We agree with the reviewer no comparison with control group is conceivable at this point. We modified the text accordingly, describing the population included and the outcomes.

2-There are paragraphs in the results section that should go in the discussion. For instance « Comparing our results… » or « As reported by Singanayagam… » and the next sentence « Xiawen et al…. » and also the sentence « Carini et al…. ».

We thank the reviewer for the comment and modified the text accordingly.

3-The results section concerning the virological results is very difficult to read. It should not repeat the data which are already in the tables. -

We edited the comments about the virological results for the sake of clarity. Thank you

4-There was one death. This is a serious adverse event. Please correct the sentence in page 6 which states that there were no adverse events.-

We agree with the reviewer and modified the text accordingly

5-There are too many details in the tables. The authors should merge table 2 and 3 .Table 5 and 6 should be merged. In tables 5 and 6, change T0, T1, etc…to the relevant days : Day 0, day 1, day 3, day 5. This will help the reader. Table 7 is not a table ; it is a figure.

We appreciate the valid suggestions and modified the tables and figure accordingly

6-Figure 1 is useless. Should be deleted.-

We deleted Figure 1

Figure 3 and Table/Figure 7 need the SD bars ->

Thank you for the suggestion. We have added SD bars to our figures

Round 2

Reviewer 3 Report

I evaluated the revised version of the manuscript by Sanduzzi Zamparelli and coll. The authors improved the manuscript. I have a few suggestions left.  

Major comments :

  • abstract :
    1. do not refer in the abstract to the comparison to literature controls since there was no formal comparison in the revised manuscript;
    2. do not conclude about efficacy since this was not assessed in the manuscript. You may suggest that controlled trials might be of interest to assess the efficacy of the compound.
    3. you may conclude about safety and feasibility of the aerosol route
  • Table 2 (line KL6) : there is a “and” in the middle of the numbers. The number of patients included should appear as a first line of the table;
  • Table 3 : the table caption should not contain any comment. The comments should be in the text of the manuscript (actually they are !). Age and BMI lines are empty and should be deleted.
  • Table 4 and Table 5 : same comment. The caption should be kept short, and any results should be given in the manuscript.
  • Figure 1 : The figure legend should be changed to be more descriptive. It could be : “ Most prevalent symptoms at covid-19 pneumonia onset. Data are expressed as the percentage of patients.
  • Figure 2. The title should be kept short. The description of the results is already in the manuscript. This repetition is unuseful.
  • The sentence “Comparing our results…were observed” should be deleted from the results section. Any discussion about the results should go in the discussion section since there was no formal comparison performed.
  • Last sentence of the results section: the authors should state simply that one patient died during the trial, due to massive pulmonary embolism. There should be no comment about the link to TAURISOLO treatment.
  • Discussion : The discussion needs to be improved. For instance, I do not understand the yellow highlighted paragraph. Is it about any effect of Taurisolo on the speed of negativation ? I would suggest the authors to be careful in their analysis of the data. They cannot write that they reached their endpoint.

Author Response

Prof. Dr. Cord Brakebusch

Editor in Chief,

Cells

On behalf of my co-authors and myself, we send to your kind attention the revised manuscript of the original study ”A Phase II study on the effect of Taurisolo® administered via AEROsol in hospitalized patients with mild to moderate COVID-19 pneumonia: the TAEROVID-19 study”  Manuscript ID: cells-1614329 for a possible publication on Cells.

The manuscript reported the results in term of feasibility and safety of Taurisolo® aerosol formulation in patients hospitalized because of Sars-CoV-2 pneumonia. This is a phase II multicentric clinical trial (TAEROVID-19) conducted during the first wave of pandemic COVID-19. we observed a rapid decline of symtomps and a low rate of intensive care in patients treated with Taurisolo®, with faster decline of symtomps. This is the first trial assessing safety and feasibility of Taurisolo® aerosol formulation. this treatment could act as an add-on therapy in the treatment of COVID 19 patients, owing to both its anti-inflammatory and antioxi-dant effects.

Compared to the first submission the manuscript has been reviewed accurately according to the valuable comments of the reviewers. As per the first major review (reviewer n 2), we added a paragraph on Taurisolo composition, mechanism of action and pharmacodynamics provided by our pharmacologist. Because of that we added his name among the coauthros (Dr. Giuseppe Annunziata- Department of Pharmacy, University Federico II, Naples, Italy; [email protected]).

Attached the point by point response to the reviewer: 

I evaluated the revised version of the manuscript by Sanduzzi Zamparelli and coll. The authors improved the manuscript. I have a few suggestions left.

Major comments :

abstract :

  1. do not refer in the abstract to the comparison to literature controls since there was no formal comparison in the revised manuscript;

              Thank you for the suggestion. We modified the text accordingly

  1. do not conclude about efficacy since this was not assessed in the manuscript. You may suggest that controlled trials might be of interest to assess the efficacy of the compound.

We appreciate the valid suggestions and modified the text accordingly

  1. you may conclude about safety and feasibility of the aerosol route

We agree with the reviewer and modified the text accordingly

Table 2 (line KL6) : there is a “and” in the middle of the numbers. The number of patients included should appear as a first line of the table;

We appreciate the valid suggestions and modified the table

Table 3 : the table caption should not contain any comment. The comments should be in the text of the manuscript (actually they are !). Age and BMI lines are empty and should be deleted. We thank the reviewer for the valuable comments. We revised the table and the caption as requested.

Table 4 and Table 5 : same comment. The caption should be kept short, and any results should be given in the manuscript.

We edited the captions of the tables. Thank you

Figure 1 : The figure legend should be changed to be more descriptive. It could be : “ Most prevalent symptoms at covid-19 pneumonia onset. Data are expressed as the percentage of patients.

We thank the reviewer for the comment and modified the text accordingly.

Figure 2. The title should be kept short. The description of the results is already in the manuscript. This repetition is unuseful.

We agree with the reviewer and modified the text accordingly

The sentence “Comparing our results…were observed” should be deleted from the results section. Any discussion about the results should go in the discussion section since there was no formal comparison performed. We agree with the reviewer; we modified the text accordingly, describing the results in the discussion section

Last sentence of the results section: the authors should state simply that one patient died during the trial, due to massive pulmonary embolism. There should be no comment about the link to TAURISOLO treatment. We thank the reviewer for the valuable comment. We removed the comment about the link with the treatment

Discussion: The discussion needs to be improved. For instance, I do not understand the yellow highlighted paragraph. Is it about any effect of Taurisolo on the speed of negativation ? I would suggest the authors to be careful in their analysis of the data. They cannot write that they reached their endpoint.

Thanks for your comment. We added a paragraph in the discussion section to discuss the data about the time of negativization in the patients treated. We agree with the reviewer that the endpoint was not reached, because only 43 patients were evaluable for analysis, a small number compared to the sample size
